# Conjugate Bayesian Chain-Rule Prediction-Powered Inference for Binary Prevalence Estimation

## Abstract

Prediction-powered inference (PPI) combines abundant machine predictions with scarce human labels to estimate population quantities with valid uncertainty. We study a focused Bayesian version of this problem: binary prevalence estimation when the operational object is a thresholded autorater decision. For an autorater decision $A \in \{0, 1\}$ and human label $H \in \{0, 1\}$, the target prevalence is

$$g = P(H = 1) = P(A = 1)P(H = 1 \mid A = 1) + P(A = 0)P(H = 1 \mid A = 0).$$

Under independent Beta priors and Bernoulli likelihoods, the posterior for the base chain-rule estimator factorizes into three independent Beta distributions, so uncertainty for $g$ is obtained by direct posterior sampling without MCMC. This conjugate base case is a specialization of broader Bayesian PPI ideas, but it gives a transparent posterior for the thresholded deployment estimand and an auditable $2 \times 2$ decision table.

We compare this Bayesian chain-rule estimator (CRE) with labeled-only Bayes, the binary difference estimator, continuous-score PPI, and a scalar power-tuned PPI baseline in the spirit of PPI++. Experiments use subject-level out-of-fold autorater probabilities from an ADNI MRI case study, with scan-level ages matched by subject identifier and acquisition date. In the full cohort (2116 scans, 503 subjects), CRE coverage is conservative-to-near-nominal in most settings, with conservative coverage in several small-budget Uniform-prior settings and mild undercoverage in a small number of Jeffreys-prior settings. Under the Uniform prior, CRE is competitive with the continuous-score baselines and generally yields shorter intervals than labeled-only Bayes and the binary difference estimator. In the scan-level age 65–70 subset (291 scans, 93 subjects), CRE remains conservative at small label budgets, whereas the binary difference estimator can under-cover despite short intervals. We also report threshold-selection audits, age-stratified discrimination, exchangeability diagnostics, conjugate K-bin score-discretization sensitivity, and simulation-based calibration. Hierarchical logit-normal and joint-threshold models are presented as non-conjugate extensions; stratified or importance-weighted variants are discussed for non-exchangeable labeled and unlabeled pools. An anonymized supplementary archive provides the analysis code, configuration files, synthetic fixtures, automated tests, and summary-level reference results; reproducing the ADNI case study additionally requires authorized access to ADNI MRI data and metadata.

## 1 Introduction

**Motivation.** High-capacity machine-learning (ML) systems are often accurate enough to be useful as large-scale autoraters, but their predictions alone do not provide valid uncertainty for population quantities when the model is biased or the deployment population differs from training data. This is especially important in biomedical monitoring, where human labels are expensive and population-level uncertainty, not only individual classification accuracy, is the inferential target.

**Prediction-powered inference and the thresholded deployment estimand.** Prediction-powered inference (PPI) uses abundant machine predictions and a smaller labeled subset to debias imputed estimates through a labeled residual or rectifier term, yielding valid confidence intervals under weak assumptions (Angelopoulos et al., 2023a). PPI++ further adapts the amount of prediction-powered correction to the quality of the predictor (Angelopoulos et al., 2023b). In this paper we focus on a related but deliberately narrower Bayesian problem. Let $A \in \{0,1\}$ denote the binary decision produced by a deployed autorater after thresholding a score, and let $H \in \{0,1\}$ denote the human label. The prevalence estimand tied to the deployed decision rule is

$$g = P(H = 1) = P(H = 1 \mid A = 1)P(A = 1) + P(H = 1 \mid A = 0)P(A = 0) = \theta_A \theta_{H|1} + (1 - \theta_A)\theta_{H|0}. \quad (1.1)$$

This thresholded estimand is distinct from continuous-score PPI, which uses probabilities $p_i$ directly. Continuous-score PPI and scalar power tuning motivated by PPI++ are strong prior-free alternatives, while CRE targets the binary operational audit table used by a deployed thresholded autorater. Both approaches are computationally lightweight; CRE's distinguishing benefit is posterior inference for the three audit-table probabilities and prior-based regularization when labeled cells are sparse, not a claim of universal efficiency dominance.

**Relationship to Bayesian PPI.** Recent Bayesian PPI work gives a general modeling framework for prediction-powered estimands, including settings with discrete autoraters and non-linear score relationships (Hofer et al., 2024). Our contribution is not to claim a wholly new Bayesian PPI paradigm. Instead, we isolate the binary chain-rule prevalence case, spell out its closed-form Beta–Bernoulli posterior, evaluate the thresholded deployment estimand against continuous-score PPI, a scalar PPI++-style power-tuned baseline, and binary baselines, and provide the practical audits needed to avoid common deployment ambiguities: subject-level out-of-fold prediction, threshold leakage, score-discretization sensitivity, exchangeability checks, and age-stratified performance.

**Contributions.** Building on classical PPI and Bayesian PPI, we make the following focused contributions:

- derive the conjugate posterior for the binary chain-rule prevalence model and propagate uncertainty to $g$ by direct Beta sampling;

- clarify the estimand distinction between thresholded CRE, the binary difference estimator, continuous-score PPI, and scalar PPI++-style power tuning;

- provide a subject-level out-of-fold ADNI MRI case study in which $H = 1$ denotes AD and $H = 0$ denotes CN, with scan-level ages matched by subject and acquisition date;

- empirically compare coverage and interval width across CRE, labeled-only Bayes, the binary difference estimator, continuous-score PPI, and scalar PPI++-style power tuning under repeated label subsampling;

- add deployment audits for threshold selection, threshold uncertainty, labeled–unlabeled exchangeability, calibration, K-bin score sensitivity, and simulation-based calibration;

- delineate when the base conjugate model is sufficient and when hierarchical pooling, importance weighting, logit-normal priors, multi-bin models with shared structure, or joint threshold uncertainty should be considered.

**Empirical overview.** Using the revised subject-level out-of-fold prediction table, the full ADNI cohort contains 2116 scans from 503 subjects, with empirical scan-level AD prevalence 0.308. The scan-level age 65–70 subset contains 291 scans from 93 subjects, with prevalence 0.237. CRE coverage is generally close to nominal: several small-budget Uniform-prior settings are conservative, while a small number of Jeffreys-prior settings mildly under-cover. CRE is best interpreted as a coherent Bayesian posterior for the thresholded deployment estimand, not as a universal replacement for continuous-score PPI.

**Limitations and scope.** The base model assumes exchangeability between the labeled subset and the autorater pool for the relevant $(A, H)$ distribution, and it conditions on a chosen threshold. It can fail under severe covariate shift, label missingness not captured by the sampling design, unstable thresholds, sparse labeled cells, or subgroup heterogeneity. Because longitudinal visits are retained, the ADNI estimand is prevalence over the assembled scan population rather than prevalence over unique participants; subject grouping prevents prediction leakage, but the base posterior does not model within-subject dependence among repeated visits. We therefore report diagnostics and sensitivity analyses, and we frame non-conjugate extensions as tools for settings in which the base conjugate assumptions are inadequate.

**Broader impact.** The method is designed for monitoring and uncertainty quantification, not for replacing clinical judgment or prospective validation. In medical imaging deployments, threshold choices and subgroup performance must be governed by domain experts and monitored prospectively.

## 2 Background and Related Work

### 2.1 Prediction–Powered inference in context: links to survey calibration, semi–supervised inference, and doubly robust estimators

Prediction–Powered Inference (PPI) exploits a high–quality but imperfect predictor trained on abundant unlabeled samples to reduce variance at fixed label budgets while preserving valid uncertainty quantification. Let $\{(X_i, Y_i)\}_{i=1}^{N}$ be inputs and outcomes and let $f$ be a predictor trained chiefly on unlabeled covariates. A purely imputed estimator $\hat{\theta}_f = N^{-1} \sum_i f(X_i)$ is generally biased, whereas the classical mean $\hat{\theta}_{\text{classical}} = N^{-1} \sum_i Y_i$ is label–hungry. PPI combines the strengths by introducing a rectifier on a small labeled subset with $n \ll N$,

$$\Delta = \frac{1}{n} \sum_{i=1}^{n} \big(f(X_i) - Y_i\big), \qquad \hat{\theta}_{\text{PPI}} = \hat{\theta}_f - \Delta,$$

and provides finite–sample guarantees under weak assumptions on the sampling of the labeled subset and mild regularity of $f$ (Angelopoulos et al., 2023a). In binary–decision settings with a machine autorater $A \in \{0, 1\}$ and a human label $H \in \{0, 1\}$, the target positivity rate $g = P(H{=}1)$ decomposes via the chain rule

$$g = P(H{=}1 \mid A{=}1)P(A{=}1) + P(H{=}1 \mid A{=}0)P(A{=}0) = \theta_A \theta_{H|1} + (1 - \theta_A)\theta_{H|0},$$

which exposes how abundant predictions (informing $\theta_A$) and scarce labels (informing $\theta_{H|a}$) jointly determine uncertainty. The same structure underlies the *difference estimator*

$$\hat{g}_{\text{diff}} = \bar{A} + \overline{(H - A)} = \frac{1}{N} \sum_{i=1}^{N} A_i + \frac{1}{n} \sum_{i=1}^{n} (H_i - A_i),$$

long studied in design–based survey sampling where auxiliary variables reduce variance without sacrificing unbiasedness (Cochran, 1977; Lohr, 2010). From this perspective, PPI can be viewed as bringing classical calibration ideas to modern ML auxiliaries. The generalized regression (GREG) and calibration estimators (Särndal et al., 1992; Deville & Särndal, 1992) adjust sampling weights (or add regression corrections) so that estimates align with known population totals; replacing "known totals" by "accurate large–$N$ machine predictions" recovers the spirit of PPI while retaining transparent conditions for coverage.

Connections to semi–supervised inference further clarify efficiency gains. A line of work on semi–supervised means, risks, and ROC–type functionals shows that plug–in estimators based on unlabeled $X$ can be augmented by labeled residuals to attain substantial variance reduction with valid inference (Chakrabortty & Cai, 2018; Gronsbell & Cai, 2018). In influence–function language, the rectifier acts as an augmentation term that centers the estimating equation, a device familiar from augmented inverse–probability weighting and targeted learning where *doubly robust* estimators remain consistent if either the outcome or propensity/auxiliary model is correct (Bang & Robins, 2005; van der Laan & Rose, 2011). PPI differs in emphasis: assumptions are simple and practically auditable (exchangeability of labeled/unlabeled pools and predictor

stability), and coverage statements leverage the large pool of machine predictions directly (Angelopoulos et al., 2023a).

For practical deployment, we emphasize two additional ingredients often underplayed in prior narratives. First, when thresholds map scores to decisions ($A = \mathbf{1}\{S \geq t\}$), *leakage* can inflate performance if the same data inform threshold selection and evaluation; out-of-fold selection and bootstrap dispersion of Youden's cutpoint mitigate this bias (Youden, 1950; Varma & Simon, 2006; Cawley & Talbot, 2010; Efron & Tibshirani, 1993). Second, because PPI and CRE hinge on exchangeability between labeled and unlabeled pools, we advocate *propensity-overlap diagnostics* (e.g., low AUC for a labeled-versus-unlabeled classifier) as a simple precondition check (Rosenbaum & Rubin, 1983).

Practical deployment also requires assessing probability calibration before scores are converted into decisions $A = \mathbf{1}\{S \geq t\}$. Post-hoc methods such as Platt or temperature scaling and isotonic regression can improve probability reliability without changing ranking and therefore do not alter threshold-agnostic discrimination (AUC) (Platt, 1999; Zadrozny & Elkan, 2002; Niculescu-Mizil & Caruana, 2005; Guo et al., 2017). In the present case study, we report reliability diagrams and Brier scores as calibration diagnostics but do not fit a post-hoc recalibration model. Continuous-score PPI and its power-tuned variant operate on $p$, whereas CRE and the binary difference estimator operate on the thresholded decision $A$.

## 2.2 Bayesian formulations of PPI, computation, and diagnostics

A Bayesian treatment of PPI provides a coherent generative specification and direct uncertainty propagation to nonlinear functionals. Hofer et al. provide a general Bayesian PPI framework and explicitly discuss discrete autoraters and non-linear score relationships (Hofer et al., 2024). Our model is a transparent binary-prevalence specialization of that broader direction, with emphasis on closed-form cell-count updates, the induced posterior for $g$, and deployment audits. With independent Beta priors and Bernoulli likelihoods for $(\theta_A, \theta_{H|1}, \theta_{H|0})$, the posterior is available without MCMC.

Some extensions break conjugacy, including hierarchical logit-normal pooling, non-Beta priors, K-bin models with shared hyperparameters, and joint inference over a data-dependent threshold. Such models could be fit with HMC or NUTS in probabilistic-programming systems (Neal, 2011; Hoffman & Gelman, 2014; Betancourt, 2017; Salvatier et al., 2016; Carpenter et al., 2017). We do not present an end-to-end NUTS experiment in this paper. If these extensions are used, standard diagnostics include rank-normalized $\hat{R}$, bulk and tail effective sample sizes, divergences, and energy diagnostics (Kumar et al., 2019; Vehtari et al., 2021; Gelman et al., 2020). Posterior predictive checks and simulation-based calibration remain useful for validating both primitive parameters and derived functionals (Gabry et al., 2019; Talts et al., 2018; McElreath, 2020).

Two modeling choices help in small–$n$ or imbalanced strata. First, weakly informative priors (e.g., Jeffreys Beta$\left(\frac{1}{2}, \frac{1}{2}\right)$) regularize extreme cell proportions when labeled counts for $A{=}1$ or $A{=}0$ are tiny, improving tail behavior of credible intervals with minimal bias (Gelman et al., 2013). Second, hierarchical extensions with partial pooling across subgroups share signal while preserving between–group differences; the induced group–wise estimands $g_g = \theta_{A,g}\theta_{H|1,g} + (1 - \theta_{A,g})\theta_{H|0,g}$ inherit shrinkage–stabilized uncertainty. Model comparison within this family can be guided by PSIS–LOO expected log predictive density to avoid over–binning or over–stratification that does not improve out–of–sample fit (Vehtari et al., 2017).

Finally, distribution shift and label–missingness mechanisms warrant explicit consideration. Under MCAR (or MAR given $A$ and coarse covariates), labeled and unlabeled pools are exchangeable for $(\theta_A, \theta_{H|a})$; sizable deviations call for stratified labeling or importance weighting under covariate–shift assumptions (Sugiyama et al., 2007). The Bayesian workflow makes such sensitivities transparent: prior–predictive checks flag implausible regions, PPCs detect lack of fit in $(A, H)$ margins, and targeted sensitivity analyses (uniform vs. Jeffreys priors; global vs. stratified thresholds) quantify the robustness of coverage and interval width—key objectives for PPI in label–scarce regimes.

## 3 Methods

**ADNI data statement (required).** Data used in this study were obtained from the Alzheimer's Disease Neuroimaging Initiative (ADNI) database (`adni.loni.usc.edu`). ADNI began in 2003 as a public–private partnership led by Michael W. Weiner, MD. Its goals include combining MRI, PET, biofluid biomarkers, and neuropsychological assessments to track progression of MCI and Alzheimer's disease, validating biomarkers for clinical trials, broadening cohort diversity, and providing data to the research community. See `adni.loni.usc.edu` for up-to-date details.

**Data version.** We downloaded ADNI data on **2025-07-14** and checked for updates prior to submission.

### 3.1 Generative specification, estimand, and extensions

Let $A \in \{0, 1\}$ denote an autorater decision (derived from a probabilistic score $p \in [0, 1]$ via a fixed operating threshold $t$) and let $H \in \{0, 1\}$ be a human label. Abundant autorater outputs are observed as $\mathcal{D}_A = \{A_i\}_{i=1}^{N_A}$ together with a comparatively small labeled subset $\mathcal{D}_H = \{(A_i, H_i)\}_{i=1}^{N_H}$, with $N_A \gg N_H$. Define

$$\theta_A = P(A = 1), \qquad \theta_{H|1} = P(H = 1 \mid A = 1), \qquad \theta_{H|0} = P(H = 1 \mid A = 0),$$

which determine the population functional

$$g(\boldsymbol{\theta}) = \theta_A \theta_{H|1} + (1 - \theta_A)\theta_{H|0}, \qquad \boldsymbol{\theta} = (\theta_A, \theta_{H|1}, \theta_{H|0}) \in (0, 1)^3. \tag{3.1}$$

For the base derivation, observational units are treated as exchangeable under the working likelihood $A_i \sim$ Bernoulli$(\theta_A)$ and $H_i \mid A_i \sim$ Bernoulli$(\theta_{H|A_i})$. The label indicator is assumed missing completely at random under the repeated-labeling design, so the labeled subset is representative of the scan population for the relevant $(A, H)$ distribution. Autorater probabilities $p_i$ are thresholded into $A_i = \mathbf{1}\{p_i \geq t\}$, where $t$ is fixed for the primary posterior; Section 3.4 audits sensitivity to data-driven threshold selection. For a step-by-step summary of how this generative specification is used in practice (from priors and likelihood to posterior summaries of $g$), see Section J.1.

**Likelihood and priors.**

**Proposition 3.1** (Conjugate posterior for the base chain-rule model). *Let* $n_A = \sum_{i=1}^{N_A} A_i$, $n_{11} = \sum_{i=1}^{N_H} \mathbf{1}\{A_i = 1, H_i = 1\}$, $n_{10} = \sum_{i=1}^{N_H} \mathbf{1}\{A_i = 1, H_i = 0\}$, $n_{01} = \sum_{i=1}^{N_H} \mathbf{1}\{A_i = 0, H_i = 1\}$, $n_{00} = \sum_{i=1}^{N_H} \mathbf{1}\{A_i = 0, H_i = 0\}$. *With independent* Beta$(\alpha_A, \beta_A)$, Beta$(\alpha_1, \beta_1)$, Beta$(\alpha_0, \beta_0)$ *priors, the posterior factorizes as*

$$\theta_A \mid \mathcal{D} \sim \text{Beta}(\alpha_A + n_A, \ \beta_A + N_A - n_A),$$

$$\theta_{H|1} \mid \mathcal{D} \sim \text{Beta}(\alpha_1 + n_{11}, \ \beta_1 + n_{10}), \quad \theta_{H|0} \mid \mathcal{D} \sim \text{Beta}(\alpha_0 + n_{01}, \ \beta_0 + n_{00}).$$

*Thus* $g = \theta_A \theta_{H|1} + (1 - \theta_A)\theta_{H|0}$ *follows by direct Monte Carlo.*

Under these assumptions, the complete–data likelihood factorizes as

$$p(\mathcal{D}_A, \mathcal{D}_H \mid \boldsymbol{\theta}) = \prod_{i=1}^{N_A} \theta_A^{A_i}(1 - \theta_A)^{1-A_i} \prod_{i=1}^{N_H} \theta_{H|1}^{H_i A_i}(1 - \theta_{H|1})^{(1-H_i)A_i} \theta_{H|0}^{H_i(1-A_i)}(1 - \theta_{H|0})^{(1-H_i)(1-A_i)}. \tag{3.2}$$

Unless stated otherwise, independent Beta$(1, 1)$ priors are placed on $(\theta_A, \theta_{H|1}, \theta_{H|0})$; Jeffreys' Beta$(\frac{1}{2}, \frac{1}{2})$ priors are used for sensitivity near the boundaries (Gelman et al., 2013). Weakly informative alternatives (e.g., logit–normal $\theta = \text{logit}^{-1}(\eta)$ with $\eta \sim \mathcal{N}(0, 1.5^2)$) are considered when extreme class imbalance or tiny stratum counts induce separation (Gelman et al., 2008).

**Identifiability and small–cell regularization.** $\theta_A$ is identified from $\mathcal{D}_A$ alone; $(\theta_{H|1}, \theta_{H|0})$ are identified from $\mathcal{D}_H$ provided each stratum $A \in \{0, 1\}$ contributes at least one labeled case asymptotically. When labeled positives (or negatives) are extremely rare in a stratum, Beta priors regularize cell–probability posteriors away from 0/1; posterior predictive checks (PPCs) on the $2 \times 2$ table clarify the degree of regularization.

**Continuous–score generalization.** If one wishes to avoid dichotomizing $p$, a $K$–bin chain rule replaces $A \in \{0,1\}$ by $B \in \{1,\dots,K\}$ with

$$g = \sum_{k=1}^{K} P(H = 1 \mid B = k)\, P(B = k).$$

with independent Beta priors on $\{P(H{=}1 \mid B = k)\}_{k=1}^{K}$ and a Dirichlet prior on $(P(B{=}1),\dots,P(B{=}K))$. This independent-bin specification remains conjugate, but it introduces more parameters and potentially sparser labeled cells; we therefore use $K = 2$ as the primary thresholded representation and assess $K \in \{4,5\}$ as a sensitivity analysis. In the full cohort, increasing $K$ changed the posterior mean of $g$ by at most $5 \times 10^{-4}$ and the interval width by at most $2.3\%$ relative to $K = 2$. In the smaller age 65–70 subset, the largest posterior-mean shift was 0.005, with widths remaining within about $3\%$ of the $K = 2$ value (Appendix H).

**Hierarchical partial pooling across strata.** For age–aware deployment with strata $s \in \{1,\dots,S\}$, parameters become $\boldsymbol{\theta}_s = (\theta_{A,s}, \theta_{H|1,s}, \theta_{H|0,s})$ and we impose logit–normal hierarchies

$$\mathrm{logit}(\theta_{H|a,s}) \sim \mathcal{N}(\mu_a, \sigma_a^2), \quad \mathrm{logit}(\theta_{A,s}) \sim \mathcal{N}(\mu_A, \sigma_A^2),$$

with weakly informative hyperpriors $\mu_{\cdot} \sim \mathcal{N}(0, 2^2)$ and $\sigma_{\cdot} \sim \mathrm{Half\text{-}Normal}(1)$, enabling partial pooling when certain strata have few labels while permitting stratum–specific deviations.

## 3.2 Bayesian computation, diagnostics, and uncertainty for functionals

Posterior inference targets $p(\boldsymbol{\theta} \mid \mathcal{D}_A, \mathcal{D}_H) \propto p(\mathcal{D}_A, \mathcal{D}_H \mid \boldsymbol{\theta})\, p(\boldsymbol{\theta})$. For the base Beta–Bernoulli model and the independent Dirichlet–Beta K-bin sensitivity model, inference is conjugate and requires no MCMC. We draw the component probabilities directly from their posterior distributions and transform each draw to $g$.

Hierarchical logit-normal models, non-Beta priors, K-bin models with shared hyperparameters, and joint threshold models would break conjugacy and could be fit with NUTS (Neal, 2011; Hoffman & Gelman, 2014; Betancourt, 2017). These models are included to delimit the scope of the base method, not as empirically validated NUTS analyses in this paper. A future NUTS implementation should report rank-normalized $\hat{R}$, bulk and tail effective sample sizes, divergences, and energy diagnostics (Vehtari et al., 2021; Gelman et al., 2020; Kumar et al., 2019).

**Posterior summaries for $g$ and contrasts.** For each direct posterior draw $\boldsymbol{\theta}^{(s)}$, we compute $g^{(s)}$ via equation 3.1 and report posterior means and equal-tailed $95\%$ credible intervals. For stratified extensions, posterior contrasts such as $g_s - g_{s'}$ would be obtained by transforming paired draws. The base analysis conditions on the fixed thresholding policy; threshold variability is assessed separately rather than folded into the primary posterior.

## 3.3 Baselines, frequentist connections, and coverage design

We compare against **four** baselines.

*(i) Labeled-only Bayes.* Model $H_i \overset{\mathrm{iid}}{\sim} \mathrm{Bernoulli}(\theta_H)$ with $\theta_H \sim \mathrm{Beta}(\alpha,\beta)$, so $g = \theta_H$ has the closed-form posterior

$$\theta_H \mid \{H_i\} \sim \mathrm{Beta}\Big(\alpha + \sum_i H_i,\ \beta + n - \sum_i H_i\Big).$$

We use the name *labeled-only Bayes* consistently to avoid the ambiguity of "naive".

*(ii) Binary difference estimator.* Define

$$\hat{g}_{\mathrm{diff}} = \bar{A} + \overline{(H - A)} = \frac{1}{N_A} \sum_{i=1}^{N_A} A_i + \frac{1}{n} \sum_{i=1}^{n} (H_i - A_i),$$

with percentile bootstrap intervals over the labeled residuals (Lohr, 2010; Efron & Tibshirani, 1993). This baseline targets the same thresholded binary decision $A$ as CRE but is prior-free and can under-cover at small label budgets.

*(iii) Prior-free analytic PPI using continuous scores.* Using probabilities $p_i$ instead of $A_i$,

$$\hat{g}_{\text{PPI}} = \bar{p} + \overline{(H - p)}, \qquad \widehat{\text{SE}}(\hat{g}_{\text{PPI}}) = \sqrt{\widehat{\text{Var}}(p)/N_A + \widehat{\text{Var}}(H - p)/n},$$

and we form intervals with a normal critical value or $t$ critical value at small $n$ (Angelopoulos et al., 2023a; Efron & Tibshirani, 1993). This is the strongest direct frequentist comparison using the continuous autorater score.

*(iv) Scalar power-tuned PPI (PPI++-style).* We also report

$$\hat{g}_\lambda = \lambda \bar{p} + \overline{(H - \lambda p)},$$

where $\lambda$ is estimated from the labeled sample to approximately minimize the scalar plug-in variance and is clipped to $[0, 1]$ (Angelopoulos et al., 2023b). We label this estimator "PPI++-style" because it implements the scalar power-tuning idea used in our notebook rather than claiming to reproduce every feature of the general PPI++ procedure.

**Repeated-labeling coverage protocol.** The primary empirical coverage study uses the revised ADNI OOF prediction table, treating the empirical scan-level prevalence in the full table as $g_{\text{true}}$ and repeatedly subsampling scan labels of sizes $n \in \{10, 20, 40, 80\}$. We use $M = 500$ repeated label subsets, posterior draws $S = 5000$ for CRE and labeled-only Bayes, and 1000 bootstrap resamples for the binary difference estimator. The same labeled subsets are reused across Uniform and Jeffreys priors so that prior sensitivity is not confounded with label-set randomness. We report empirical coverage and mean 95% interval width. This experiment evaluates the scan-level estimand and does not provide a cluster-robust subject-level coverage guarantee.

**Asymptotic connection to the difference estimator.** CRE and the binary difference estimator use the same thresholded decision table but differ in uncertainty representation. The CRE posterior mean is a regularized chain-rule plug-in estimate, while the difference estimator is a design-based residual correction. Appendix A gives a conservative algebraic comparison and removes the stronger $O_{\mathbb{P}}(N_A^{-1})$ claim from the previous version.

### 3.4 Operating thresholds, calibration assessment, and threshold sensitivity

Autorater scores $p$ are mapped to hard decisions $A = \mathbf{1}\{p \geq t\}$.

**Selection of $t$ (OOF vs. leaky).** We study the prespecified deployment threshold $t = 0.5$ and Youden's descriptive threshold $t_Y^\star \in \arg\max_t\{\text{TPR}(t) + \text{TNR}(t) - 1\}$ (Youden, 1950; Fluss et al., 2005). The CNN probabilities are first generated by five-fold subject-grouped out-of-fold prediction, with all scans from a participant assigned to the same fold. Separately, for the threshold-selection audit, we partition these fixed OOF predictions by subject, select Youden's threshold using the training subjects, and evaluate that threshold on held-out subjects. No subject is shared between threshold selection and evaluation. We also report a same-scope, full-data Youden threshold explicitly as a descriptive leaky benchmark rather than as an unbiased performance estimate.

**Propagation into CRE.** For any prespecified threshold $t$, one can recompute $A$ and refit CRE so that the conditional probabilities correspond to that operating rule. In the primary analysis we fix $t = 0.5$. We bootstrap the full OOF table to quantify the dispersion of the descriptive Youden threshold and the resulting ACC, TPR, and TNR. These threshold results are an operating-point sensitivity analysis; the reported CRE intervals remain conditional on the fixed primary threshold rather than integrating over a threshold posterior.

**Calibration assessment.** We assess probability reliability using calibration curves and Brier scores (Brier, 1950; Murphy, 1973). We do not fit temperature scaling or isotonic regression in the present analysis. In a prospective deployment, such a transformation could be estimated on separate development data before fixing the operational threshold (Guo et al., 2017; Zadrozny & Elkan, 2002).

### 3.5 Case-study pipeline: data curation, autorater, and end-to-end flow

The medical-imaging case study uses ADNI T1-weighted MRI scans restricted to AD and CN labels; $H = 1$ denotes AD and $H = 0$ denotes CN. DICOM-to-NIfTI conversion uses `dcm2niix`; volumes are loaded with NiBabel (Li et al., 2016; Brett et al., 2020). Preprocessing includes center crop/pad to a brain bounding box, resampling to $64^3$, and per-volume normalization. The autorater is a lightweight 3D CNN trained with Adam at $10^{-4}$, batch size 2, for 5 epochs in PyTorch/`numpy` (Kingma & Ba, 2014; He et al., 2015; Paszke et al., 2019; Harris et al., 2020).

To address leakage concerns, all autorater probabilities used in the inference experiments are subject-level out-of-fold predictions. All scans from the same ADNI participant are assigned to the same fold, and no subject appears in more than one OOF fold. The final prediction table contains 2116 scans from 503 subjects; the OOF fold counts are 424, 440, 420, 405, and 427 scans. Scan-level age is matched by ADNI subject identifier and acquisition date, yielding no missing ages. The exact age 65–70 subset contains 291 scans from 93 subjects.

Longitudinal visits are retained, so the empirical target in this case study is scan-level AD-label prevalence over the assembled scan population. Subject grouping is used for model training and evaluation to prevent within-subject prediction leakage, while the base prevalence posterior treats scans as the observational units and does not include a longitudinal random effect. Scores $p_i$ are thresholded into $A_i = \mathbf{1}\{p_i \geq 0.5\}$ for the primary deployment estimand. We additionally report Youden-threshold analyses as operating-point audits, not as the primary estimand. The end-to-end routine follows Section J.2 for threshold auditing and Section J.1 for conjugate Bayesian inference.

**Robustness to dataset shift and label missingness.** If labeled cases differ from the unlabeled pool, the exchangeability assumption fails. Practical responses include stratified label acquisition, labeled-versus-unlabeled propensity diagnostics, and explicit selection or missingness models (Rosenbaum & Rubin, 1983; Sugiyama et al., 2007). Appendix L outlines an importance-weighted generalized-Bayes pseudo-posterior based on density-ratio-weighted sufficient statistics. This is a sensitivity construction, not an empirical claim established by the present ADNI experiment.

Table 1: Practical guide for choosing an inference strategy.

| Deployment condition | Suggested analysis |
|---|---|
| Fixed binary decision and exchangeable random labels | Base conjugate CRE |
| Continuous score is the inferential auxiliary | Continuous-score PPI or scalar power-tuned PPI |
| Adequate labels within score regions | Conjugate independent-bin K-bin sensitivity analysis |
| Sparse heterogeneous subgroups | Hierarchical partial pooling |
| Labeled/unlabeled covariate shift | Stratified labeling or importance weighting |
| Threshold is repeatedly re-selected | Subject-grouped threshold audit, bootstrap sensitivity, or a joint-threshold model |
| Nonrandom label missingness | Explicit missingness or selection model |
| Repeated visits are the inferential unit | Cluster-aware or longitudinal extension |

# 4 Experiments

## 4.1 Repeated-labeling coverage and posterior behavior

The revised coverage experiment uses subject-level out-of-fold autorater probabilities from the ADNI prediction table. The full cohort contains 2116 scans from 503 subjects with empirical AD prevalence $g_{\text{true}} = 0.3081$. The scan-level age 65–70 subset contains 291 scans from 93 subjects with empirical prevalence $g_{\text{true}} = 0.2371$. No subject appears in multiple OOF folds. At the fixed threshold $t = 0.5$, the full cohort has $P(A = 1) = 0.1418$, AUC 0.6880, accuracy 0.7136, TPR 0.2653, and TNR 0.9133. The age 65–70 subset has $P(A = 1) = 0.1203$, AUC 0.7717, accuracy 0.8351, TPR 0.4058, and TNR 0.9685.

Table 2 reports the full-cohort repeated-labeling results under the Uniform prior. CRE coverage ranges from 0.944 to 0.970 and its mean interval width decreases from 0.449 at $n = 10$ to 0.189 at $n = 80$. In this experiment CRE is competitive with continuous-score PPI and the scalar power-tuned PPI baseline, while generally yielding shorter intervals than labeled-only Bayes and the binary difference estimator. We do not claim uniform dominance: the continuous-score methods retain information discarded by thresholding, whereas CRE provides a posterior for the operational binary audit table.

Table 2: Full cohort repeated-labeling coverage and mean 95% interval width using subject-level OOF autorater probabilities. CRE uses the thresholded decision $A = \mathbf{1}\{p \geq 0.5\}$; PPI and the PPI++-style power-tuned baseline use continuous probabilities. Uniform prior results are shown; Jeffreys sensitivity appears in the appendix.

| $n$ | CRE | | Labeled-only Bayes | | Binary Diff. | | PPI | | PPI++-style | |
|---|---|---|---|---|---|---|---|---|---|---|
| | Cov. | W | Cov. | W | Cov. | W | Cov. | W | Cov. | W |
| 10 | 0.970 | 0.449 | 0.946 | 0.482 | 0.886 | 0.570 | 0.948 | 0.626 | 0.938 | 0.585 |
| 20 | 0.964 | 0.345 | 0.960 | 0.369 | 0.930 | 0.419 | 0.948 | 0.413 | 0.942 | 0.396 |
| 40 | 0.944 | 0.259 | 0.942 | 0.272 | 0.942 | 0.309 | 0.940 | 0.277 | 0.918 | 0.268 |
| 80 | 0.962 | 0.189 | 0.952 | 0.197 | 0.946 | 0.220 | 0.950 | 0.197 | 0.948 | 0.191 |

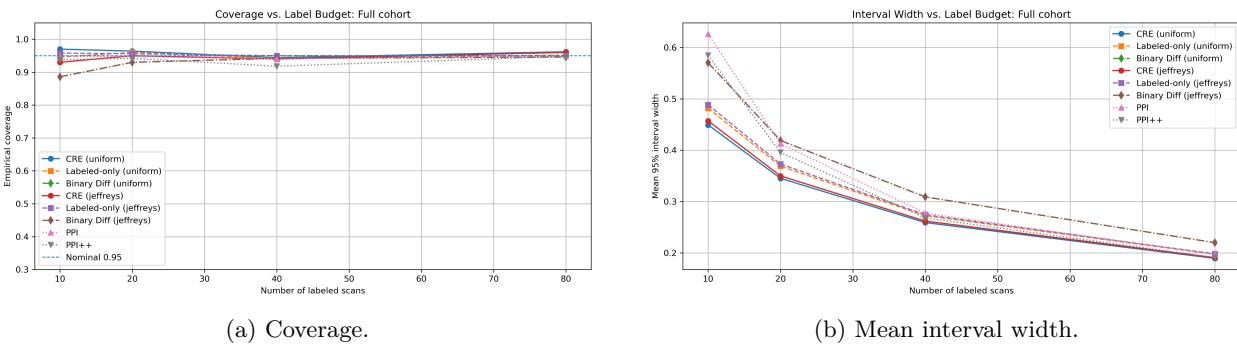

(a) Coverage.  (b) Mean interval width.

Figure 1: Full-cohort repeated-labeling coverage and width using subject-level OOF predictions.

For the smaller age 65–70 subset, CRE remains conservative under the Uniform prior: coverage/width are $0.980/0.414$, $0.982/0.306$, $0.978/0.228$, and $0.986/0.166$ for $n = 10, 20, 40, 80$, respectively. At $n = 10$, the binary difference estimator has shorter intervals but severe under-coverage (0.778), illustrating the coverage-width tradeoff that motivated the Bayesian regularization.

## 4.2 ADNI case study: discrimination, thresholds, calibration, and subgroup analysis

Table 3 summarizes discrimination under the revised subject-level OOF predictions. The OOF AUC is moderate overall and varies by age stratum. A pairwise permutation test with Holm correction finds a significant AUC difference between the 50–73 and 80–100 strata ($p_{\mathrm{Holm}} = 0.0138$), while the other pairwise differences are not significant after correction (Appendix K).

One possible explanation for the lower AUC in the 80–100 stratum is greater anatomical and clinical heterogeneity among the oldest scans, including age-related changes that may reduce separation between AD and CN scores. We did not directly test this mechanism, so we treat it as a hypothesis rather than a causal conclusion. The result indicates weaker autorater discrimination and calibration in this stratum, not an algebraic failure of the CRE posterior. A weaker autorater can reduce the efficiency gains available from prediction-powered inference and should therefore be reflected in deployment monitoring. Consistent with this interpretation, the 80–100 stratum also has the largest Brier score among the prespecified age bins.

Table 3: Subject-level OOF discrimination and fixed-threshold performance. The table reports the AUC interval produced by the revised analysis and point estimates of ACC, TPR, and TNR.

| Scope | $N$ | Subjects | Prev. | AUC | 95% AUC CI | ACC | TPR | TNR |
|---|---|---|---|---|---|---|---|---|
| Overall | 2116 | 503 | 0.308 | 0.688 | [0.664, 0.713] | 0.714 | 0.265 | 0.913 |
| 50–73 | 737 | 205 | 0.294 | 0.722 | [0.682, 0.762] | 0.757 | 0.267 | 0.962 |
| 74–79 | 695 | 218 | 0.291 | 0.700 | [0.655, 0.743] | 0.709 | 0.238 | 0.903 |
| 80–100 | 684 | 204 | 0.341 | 0.638 | [0.593, 0.683] | 0.671 | 0.288 | 0.869 |
| 65–70 | 291 | 93 | 0.237 | 0.772 | [0.699, 0.841] | 0.835 | 0.406 | 0.968 |

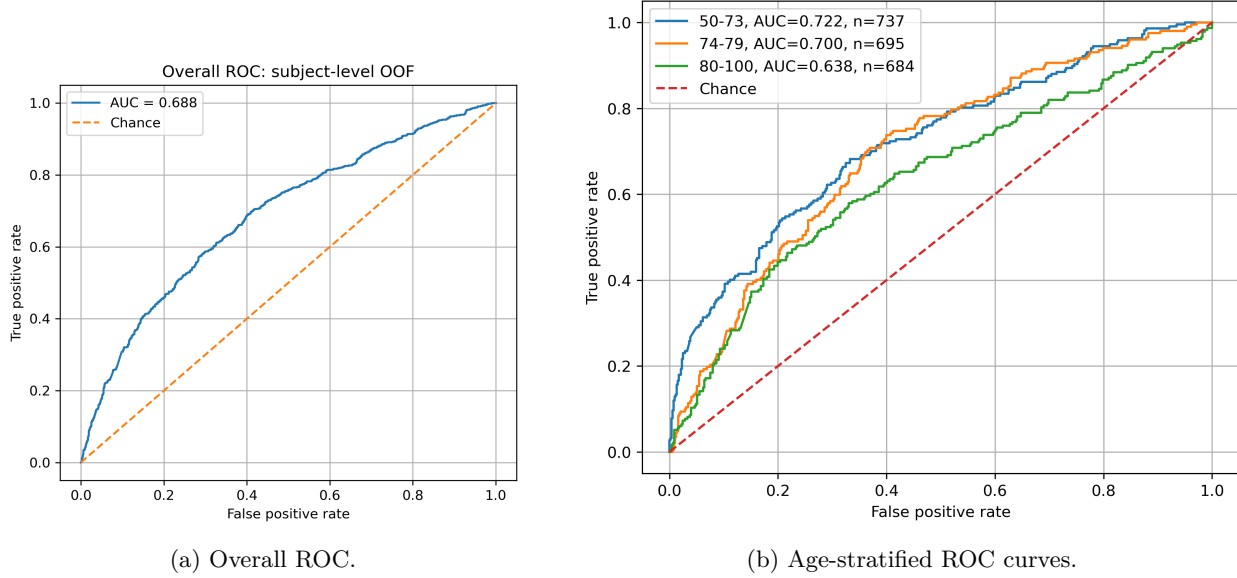

(a) Overall ROC.

(b) Age-stratified ROC curves.

Figure 2: Subject-level OOF ROC curves.

Table 4 shows that the fixed 0.5 threshold is specificity-heavy. Youden-selected thresholds increase sensitivity but reduce specificity and overall accuracy. We therefore use $t = 0.5$ for the primary CRE deployment estimand and treat Youden analyses as operating-point audits.

A subject-grouped OOF threshold-selection audit produced training thresholds in the range 0.206–0.290, with zero subject overlap between threshold-selection and held-out evaluation folds. Bootstrap resampling showed

Table 4: Fixed-threshold and same-scope Youden operating points. Youden rows are descriptive audits and are not used as the primary fixed-threshold estimand.

| Scope | Threshold policy | $t$ | ACC | TPR | TNR |
|---|---|---|---|---|---|
| Overall | fixed 0.5 | 0.500 | 0.714 | 0.265 | 0.913 |
| Overall | Youden descriptive | 0.212 | 0.627 | 0.689 | 0.600 |
| 50–73 | fixed 0.5 | 0.500 | 0.757 | 0.267 | 0.962 |
| 50–73 | Youden descriptive | 0.204 | 0.672 | 0.682 | 0.667 |
| 74–79 | fixed 0.5 | 0.500 | 0.709 | 0.238 | 0.903 |
| 74–79 | Youden descriptive | 0.203 | 0.653 | 0.708 | 0.631 |
| 80–100 | fixed 0.5 | 0.500 | 0.671 | 0.288 | 0.869 |
| 80–100 | Youden descriptive | 0.345 | 0.639 | 0.579 | 0.670 |
| 65–70 | fixed 0.5 | 0.500 | 0.835 | 0.406 | 0.968 |
| 65–70 | Youden descriptive | 0.388 | 0.821 | 0.565 | 0.901 |

non-negligible Youden-threshold dispersion: mean threshold 0.251 with 95% bootstrap range $[0.187, 0.344]$. This supports reporting data-driven thresholds as sensitivity analyses rather than silently substituting them for the primary fixed threshold.

**Prevalence estimation on real data.** Using all available labels descriptively, PPI, the PPI++-style power-tuned baseline, and CRE agree closely overall and by age (Table 5). This table is descriptive and distinct from the repeated-labeling coverage study.

Table 5: Descriptive prevalence estimates using all available labels. Intervals are 95% intervals.

| Scope | PPI | Power-tuned PPI | CRE Uniform | CRE Jeffreys |
|---|---|---|---|---|
| Overall | 0.308 [0.287, 0.330] | 0.308 [0.289, 0.327] | 0.309 [0.289, 0.329] | 0.308 [0.289, 0.328] |
| 50–73 | 0.294 [0.260, 0.329] | 0.294 [0.263, 0.326] | 0.295 [0.262, 0.328] | 0.295 [0.263, 0.328] |
| 74–79 | 0.291 [0.254, 0.328] | 0.291 [0.258, 0.324] | 0.291 [0.258, 0.326] | 0.291 [0.258, 0.325] |
| 80–100 | 0.341 [0.301, 0.381] | 0.341 [0.306, 0.376] | 0.341 [0.306, 0.376] | 0.341 [0.307, 0.376] |
| 65–70 | 0.237 [0.187, 0.287] | 0.237 [0.191, 0.283] | 0.240 [0.192, 0.289] | 0.238 [0.191, 0.288] |

**Calibration and exchangeability audits.** The Brier scores show the same age trend: 0.202 overall, 0.182 for ages 50–73, 0.196 for 74–79, 0.229 for 80–100, and 0.137 for the age 65–70 subset. As a sanity check for labeled–unlabeled exchangeability under random label subsampling, a propensity classifier distinguishing pseudo-labeled from unlabeled rows had mean AUC 0.504 across 50 repetitions, close to chance.

### 4.3 Computation, diagnostics, and reproducibility

The base CRE and labeled-only Bayes computations use direct conjugate sampling and therefore require no MCMC. In the revised coverage experiments we used 5000 posterior draws for each repeated label subset. SBC directly checks the conjugate implementation for the target functional $g$. The non-conjugate models discussed in Methods were not run end-to-end; a future implementation should report the usual $\hat{R}$, bulk/tail ESS, divergence, and energy diagnostics.

**Summary.** Across the revised subject-level OOF experiments, CRE provides conservative-to-near-nominal coverage for the thresholded deployment estimand in most settings, with conservative small-budget behavior under several Uniform-prior settings and mild undercoverage in a small number of Jeffreys-prior settings. It is competitive with the continuous-score baselines, exposes threshold sensitivity transparently, and yields stable prevalence estimates under conjugate K-bin score discretization.

## 5 Conclusion

We presented a focused Bayesian chain-rule version of prediction-powered inference for binary prevalence estimation with a thresholded autorater. In the base Beta–Bernoulli specification, the posterior factorizes into three independent Beta distributions, so uncertainty for $g = \theta_A \theta_{H|1} + (1 - \theta_A)\theta_{H|0}$ is propagated by direct sampling rather than MCMC. The revised framing is intentionally narrow: CRE is a coherent Bayesian posterior for the thresholded deployment estimand, not a universal replacement for continuous-score PPI.

The revised ADNI analysis uses subject-level out-of-fold autorater probabilities, scan-level ages matched by subject identifier and acquisition date, and an explicit binary label definition with $H = 1$ for AD and $H = 0$ for CN. Continuous-score PPI and the scalar power-tuned PPI baseline remain strong comparators. CRE's practical value is not universal interval-width dominance, but a directly interpretable posterior over the thresholded $2 \times 2$ decision table used in deployment.

The binary chain-rule model is a specialization of broader Bayesian PPI work that already includes discrete autoraters. Our contribution is the explicit conjugate prevalence specialization, the coverage-width evaluation against binary and continuous-score baselines, and the deployment-audit workflow. The independent-bin K-bin sensitivity remains conjugate. Hierarchical logit-normal and joint-threshold models are non-conjugate future extensions, while stratified or importance-weighted analyses address possible non-exchangeability. Future work should also develop cluster-aware inference for longitudinal scans and evaluate non-conjugate extensions with full sampler diagnostics.

**Use of large language models.** During manuscript revision, we used ChatGPT to assist with code organization, debugging, language editing, and structuring responses to reviewer comments. All mathematical derivations, experimental choices, code outputs, references, and manuscript claims were reviewed and verified by the authors, who take full responsibility for the final content.

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

## A  Theory snap-in: relationship between CRE and the binary difference estimator

The previous version stated an $O_{\mathbb{P}}(N_A^{-1})$ equivalence between the CRE plug-in and the binary difference estimator. That rate is not generally valid because the discrepancy depends on the labeled-subset autorater proportion.

Let

$$\bar{A}_{\text{all}} = \frac{1}{N_A}\sum_{i=1}^{N_A} A_i, \qquad \bar{A}_{\text{lab}} = \frac{1}{N_H}\sum_{i\in\mathcal{L}} A_i,$$

and, assuming both labeled autorater strata are nonempty, define

$$\hat{\theta}_{H|1} = \frac{\sum_{i\in\mathcal{L}} A_i H_i}{\sum_{i\in\mathcal{L}} A_i}, \qquad \hat{\theta}_{H|0} = \frac{\sum_{i\in\mathcal{L}}(1-A_i)H_i}{\sum_{i\in\mathcal{L}}(1-A_i)}.$$

The chain-rule plug-in estimator is

$$\hat{g}_{\text{CR,plug}} = \bar{A}_{\text{all}}\hat{\theta}_{H|1} + (1-\bar{A}_{\text{all}})\hat{\theta}_{H|0}.$$

Because

$$\bar{H}_{\text{lab}} = \bar{A}_{\text{lab}}\hat{\theta}_{H|1} + (1-\bar{A}_{\text{lab}})\hat{\theta}_{H|0},$$

the difference estimator obeys the exact identity

$$
\begin{aligned}
\hat{g}_{\text{diff}} - \hat{g}_{\text{CR,plug}} = (\bar{A}_{\text{all}} - \bar{A}_{\text{lab}}) \\
\times \big(1 - \hat{\theta}_{H|1} + \hat{\theta}_{H|0}\big).
\end{aligned}
\tag{A.1}
$$

Thus the finite-sample gap is controlled by the difference between the full-pool and labeled-subset autorater proportions. Under an i.i.d. superpopulation formulation with exchangeable random labeling and nondegenerate strata, the gap is generally $O_{\mathbb{P}}(N_H^{-1/2} + N_A^{-1/2})$, and is dominated by $N_H^{-1/2}$ when $N_H \ll N_A$.

A first-order expansion of $g(\theta_A, \theta_{H|1}, \theta_{H|0})$ around the empirical plug-in values gives

$$
\mathbb{E}[g \mid \mathcal{D}] \approx g(\hat{\boldsymbol{\theta}}) + (\hat{\theta}_{H|1} - \hat{\theta}_{H|0})\{\mathbb{E}(\theta_A \mid \mathcal{D}) - \hat{\theta}_A\} + \hat{\theta}_A\{\mathbb{E}(\theta_{H|1} \mid \mathcal{D}) - \hat{\theta}_{H|1}\} + (1 - \hat{\theta}_A)\{\mathbb{E}(\theta_{H|0} \mid \mathcal{D}) - \hat{\theta}_{H|0}\}.
$$

Thus the CRE posterior mean can be viewed as a Beta-regularized chain-rule plug-in estimate. The regularization is most visible when labeled cells are sparse or imbalanced, which explains why CRE can be more stable than the binary difference estimator at small label budgets.

## B    Prior-free PPI baseline (analytic): estimator and CIs

For the prevalence functional, the *prediction–powered* estimator using probabilities is

$$
\hat{g}_{\text{PPI}} = \bar{p} + \overline{(H - p)} = \frac{1}{N_A}\sum_{i=1}^{N_A} p_i + \frac{1}{N_H}\sum_{i=1}^{N_H}(H_i - p_i).
$$

with large-sample variance $\widehat{\text{Var}}(\hat{g}_{\text{PPI}}) = \widehat{\text{Var}}(p)/N_A + \widehat{\text{Var}}(H-p)/N_H$. For very small $n$ we use a $t$-critical value with Satterthwaite's degrees of freedom; otherwise $z_{0.975}$ suffices. **This is distinct from the *difference estimator*** based on binary $A$, $\hat{g}_{\text{diff}} = \bar{A} + \overline{(H - A)}$, for which we report nonparametric bootstrap percentile CIs (see Methods).

## C    Exchangeability diagnostics (propensity overlap)

As a sanity check for the random label-subsampling design used in the repeated-labeling experiments, we repeatedly marked 30% of rows as pseudo-labeled and trained a logistic classifier to distinguish pseudo-labeled from unlabeled rows using age and autorater score. Across 50 repetitions, the mean propensity AUC was 0.504 with empirical 2.5% and 97.5% quantiles 0.463 and 0.538, close to chance. This does not prove exchangeability in arbitrary deployments; it verifies the random subsampling design used in our experiments and provides a template for future deployment audits.

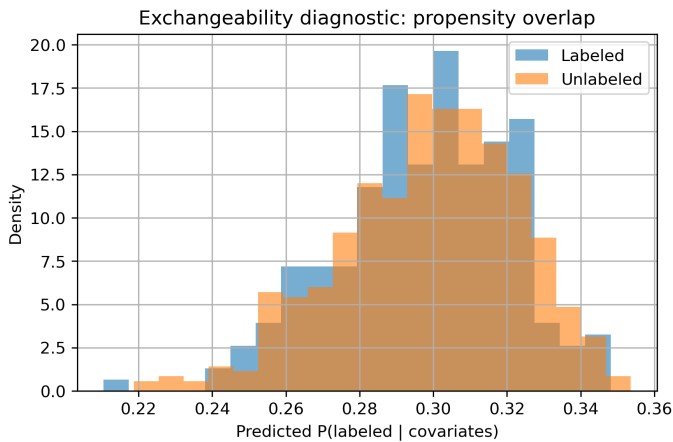

Figure 3: Propensity overlap histogram for pseudo-labeled vs. unlabeled rows.

## D  Threshold selection: OOF vs. leaky and bootstrap dispersion

Subject-grouped OOF threshold selection chose Youden thresholds on training subjects and evaluated them on held-out subjects. No subject overlapped between threshold-selection and evaluation folds.

Table 6: Subject-grouped OOF threshold audit.

| Fold | $t_{\mathrm{train}}$ | ACC | TPR | TNR | Subject overlap |
|---|---|---|---|---|---|
| 1 | 0.290 | 0.749 | 0.698 | 0.770 | 0 |
| 2 | 0.286 | 0.687 | 0.602 | 0.723 | 0 |
| 3 | 0.206 | 0.570 | 0.560 | 0.575 | 0 |
| 4 | 0.280 | 0.612 | 0.590 | 0.618 | 0 |
| 5 | 0.261 | 0.610 | 0.542 | 0.661 | 0 |

Bootstrap resampling of the full table produced mean Youden threshold 0.251 and 95% bootstrap range $[0.187, 0.344]$. The induced operating-point ranges were ACC $[0.605, 0.687]$, TPR $[0.497, 0.727]$, and TNR $[0.551, 0.772]$.

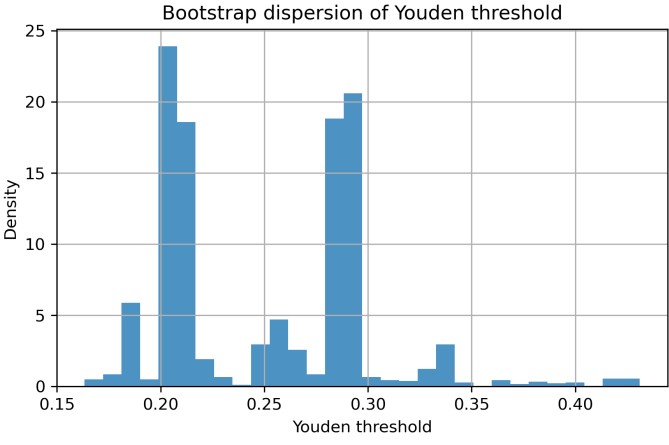

Figure 4: Bootstrap distribution of Youden thresholds.

## E   DE and PPI baselines: additional simulation tables

Detailed DE vs. CRE numbers are reported in the main text (Table 2); extended tables and per-replication summaries are included in the repository.

## F   Additional results: Age 65–70 subset

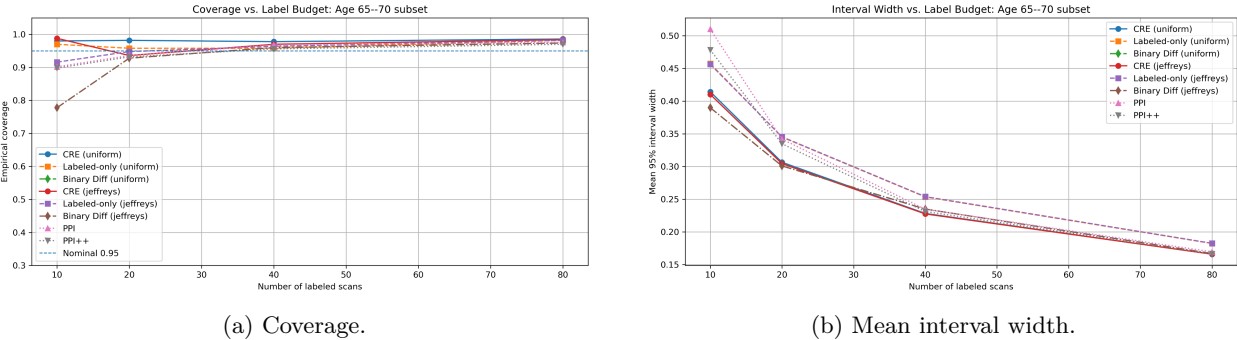

(a) Coverage.                                               (b) Mean interval width.

Figure 5: Scan-level age 65–70 subset: repeated-labeling coverage and width using subject-level OOF predictions.

Table 7: Age 65–70 subset: empirical 95% coverage and mean interval width under the Uniform prior.

| $n$ | CRE | | Labeled-only Bayes | | Binary Diff. | | PPI | | PPI++-style | |
|---|---|---|---|---|---|---|---|---|---|---|
| | Cov. | W | Cov. | W | Cov. | W | Cov. | W | Cov. | W |
| 10 | 0.980 | 0.414 | 0.970 | 0.457 | 0.778 | 0.390 | 0.902 | 0.510 | 0.898 | 0.478 |
| 20 | 0.982 | 0.306 | 0.958 | 0.345 | 0.928 | 0.301 | 0.936 | 0.342 | 0.932 | 0.335 |
| 40 | 0.978 | 0.228 | 0.958 | 0.254 | 0.960 | 0.235 | 0.968 | 0.235 | 0.956 | 0.231 |
| 80 | 0.986 | 0.166 | 0.982 | 0.182 | 0.974 | 0.167 | 0.976 | 0.170 | 0.970 | 0.166 |

## G   Prevalence estimates on real data (overall and by age)

The main text reports the descriptive all-label prevalence estimates for PPI, the PPI++-style power-tuned baseline, CRE Uniform, and CRE Jeffreys. These estimates are not repeated-labeling coverage estimates; they summarize agreement among methods when all available labels are used.

## H   K-bin sensitivity (quantile binning)

For the $K$-bin sensitivity analysis, we partitioned subject-level OOF autorater probabilities into $K \in \{2, 4, 5\}$ quantile bins and estimated

$$g = \sum_{k=1}^{K} P(H = 1 \mid B = k)P(B = k).$$

We used a Dirichlet prior for the bin probabilities and independent Beta priors for the conditional label probabilities within each bin. This independent-bin specification remains conjugate, so it is a score-discretization sensitivity analysis rather than a non-conjugate NUTS experiment.

Table 8: K-bin CRE sensitivity using subject-level OOF autorater scores. Increasing the number of score bins has little effect on the posterior mean or interval width.

| Dataset | Prior | $K$ | Mean | 95% CI | Width | $\Delta$ mean | Width ratio |
|---------|-------|-----|------|--------|-------|---------------|-------------|
| Full | Uniform | 2 | 0.3086 | [0.2890, 0.3287] | 0.0398 | 0.0000 | 1.0000 |
| Full | Uniform | 4 | 0.3090 | [0.2896, 0.3288] | 0.0392 | 0.0005 | 0.9847 |
| Full | Uniform | 5 | 0.3091 | [0.2895, 0.3288] | 0.0393 | 0.0005 | 0.9880 |
| Full | Jeffreys | 2 | 0.3083 | [0.2885, 0.3276] | 0.0391 | 0.0000 | 1.0000 |
| Full | Jeffreys | 4 | 0.3086 | [0.2889, 0.3285] | 0.0395 | 0.0004 | 1.0124 |
| Full | Jeffreys | 5 | 0.3085 | [0.2886, 0.3285] | 0.0400 | 0.0003 | 1.0232 |
| 65–70 | Uniform | 2 | 0.2408 | [0.1928, 0.2925] | 0.0997 | 0.0000 | 1.0000 |
| 65–70 | Uniform | 4 | 0.2443 | [0.1976, 0.2948] | 0.0972 | 0.0035 | 0.9756 |
| 65–70 | Uniform | 5 | 0.2458 | [0.1991, 0.2957] | 0.0966 | 0.0050 | 0.9692 |
| 65–70 | Jeffreys | 2 | 0.2389 | [0.1919, 0.2895] | 0.0976 | 0.0000 | 1.0000 |
| 65–70 | Jeffreys | 4 | 0.2407 | [0.1936, 0.2906] | 0.0970 | 0.0019 | 0.9942 |
| 65–70 | Jeffreys | 5 | 0.2415 | [0.1950, 0.2912] | 0.0962 | 0.0027 | 0.9859 |

## I  SBC and additional diagnostics

Simulation-based calibration (SBC) was used as a computational check for the conjugate posterior sampler. We simulated $M = 500$ datasets with $S = 1000$ posterior draws per dataset, $N_A = 2116$, and $N_H = 100$. For the derived prevalence functional $g$, rank histograms were consistent with uniformity under both Jeffreys and Uniform priors. Jeffreys-prior primitive parameters also showed no evidence of rank non-uniformity. Under the Uniform prior, $\theta_A$ and, marginally, $\theta_{H|0}$ showed small deviations in omnibus tests, so we report the full diagnostics and use SBC primarily as a computational check for the target functional $g$.

Table 9: SBC rank-uniformity diagnostics for the conjugate CRE sampler. We report $\chi^2$ and KS test $p$-values for posterior ranks across $M = 500$ simulated datasets with $S = 1000$ posterior draws. Expected count per bin is 25.

| Prior | Parameter | $\chi^2 p$ | KS $p$ | Min bin | Max bin |
|-------|-----------|------------|--------|---------|---------|
| Jeffreys | $\theta_A$ | 0.225 | 0.143 | 16 | 37 |
| Jeffreys | $\theta_{H|1}$ | 0.710 | 0.539 | 18 | 38 |
| Jeffreys | $\theta_{H|0}$ | 0.945 | 0.678 | 20 | 34 |
| Jeffreys | $g$ | 0.593 | 0.229 | 13 | 32 |
| Uniform | $\theta_A$ | 0.016 | 0.006 | 14 | 38 |
| Uniform | $\theta_{H|1}$ | 0.652 | 0.843 | 19 | 34 |
| Uniform | $\theta_{H|0}$ | 0.317 | 0.047 | 14 | 36 |
| Uniform | $g$ | 0.923 | 0.612 | 19 | 35 |

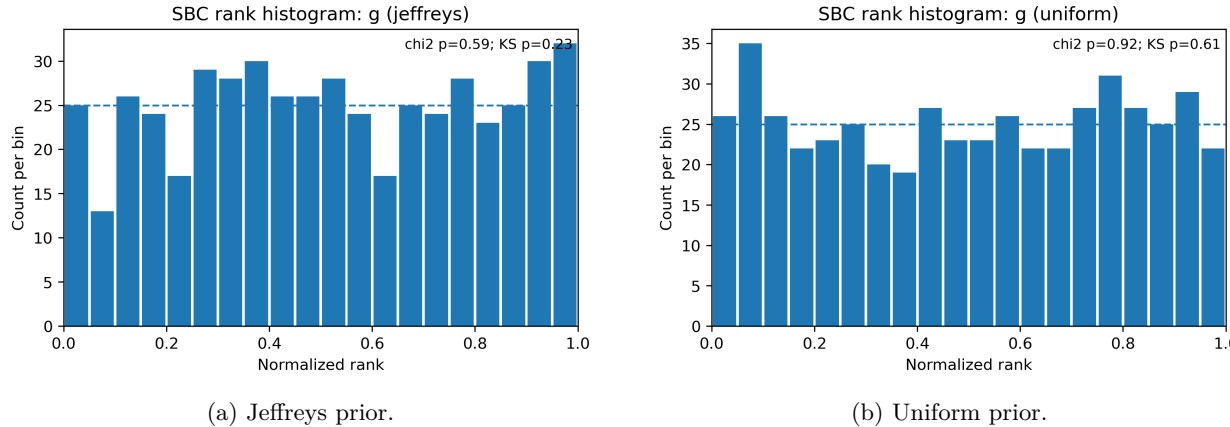

(a) Jeffreys prior.            (b) Uniform prior.

Figure 6: SBC rank histograms for the derived prevalence functional $g$.

## J   Algorithms (pseudocode)

### J.1   Bayesian CRE inference

---
**Algorithm 1** Conjugate binary CRE

---
**Require:** $\mathcal{D}_A = \{A_i\}_{i=1}^{N_A}$, $\mathcal{D}_H = \{(A_i, H_i)\}_{i=1}^{N_H}$, independent Beta priors
 1: Compute $(n_A, n_{11}, n_{10}, n_{01}, n_{00})$
 2: Update the three Beta posterior shapes using Proposition 3.1
 3: **for** $s = 1, \ldots, S$ **do**
 4:  Draw $\theta_A^{(s)}, \theta_{H|1}^{(s)}, \theta_{H|0}^{(s)}$ independently from their Beta posteriors
 5:  Set $g^{(s)} = \theta_A^{(s)}\theta_{H|1}^{(s)} + (1 - \theta_A^{(s)})\theta_{H|0}^{(s)}$
 6: **end for**
 7: Report posterior summaries of $g$ and the three auditable cell probabilities

---

Non-conjugate hierarchical, shared-hyperparameter K-bin, or joint-threshold extensions require a separate posterior sampler. If NUTS is used, $\hat{R}$, bulk/tail ESS, divergences, and energy diagnostics should be reported; no such end-to-end extension is claimed in the present experiments.

### J.2   Operating-threshold audit

---
**Algorithm 2** Threshold and calibration-assessment workflow

---
**Require:** Scores $p_i$, labels $H_i$ (dev split), policy $\in \{t{=}0.5, t_Y^\star\}$
 1: **if** $t_Y^\star$ **then**
 2:  Sweep $t$; pick $t$ maximizing $J(t) = \mathrm{TPR}(t) + \mathrm{TNR}(t) - 1$
 3:  Bootstrap labeled set $B$ times to obtain $\{t_{Y,b}^\star\}$ and summarize
 4: **end if**
 5: Assess calibration using reliability curves and Brier scores
 6: If recalibration is part of a future deployment, fit it on separate development data
 7: Apply the prespecified $t$ to obtain $A$
 8: Fit CRE on $(\mathcal{D}_A, \mathcal{D}_H)$; report the posterior for $g$

---

## K   Permutation tests: age-stratum AUC comparisons

Holm-adjusted $p$-values are based on two-sided permutation tests comparing age-bin AUCs under subject-level OOF predictions.

Table 10: Pairwise AUC differences across age strata.

| Comparison | AUC diff | Raw $p$ | Holm $p$ |
|---|---|---|---|
| 50–73 vs. 74–79 | 0.0223 | 0.4480 | 0.4480 |
| 50–73 vs. 80–100 | 0.0836 | 0.0046 | 0.0138 |
| 74–79 vs. 80–100 | 0.0613 | 0.0550 | 0.1100 |

## L   Importance-weighted CRE under covariate shift

Suppose unlabeled data follow $p_{\text{pop}}(X)$ while labels come from $p_{\text{lab}}(X)$. Let stabilized importance weights $w(x) \propto p_{\text{pop}}(x)/p_{\text{lab}}(x)$, and normalized $\tilde{w}_i = w(X_i)/\left(\frac{1}{n}\sum_j w(X_j)\right)$. Replace unweighted $(A, H)$ margins by their weighted analogs; for the unlabeled pool use $\tilde{v}_i$ analogously. Kish effective sizes $n_{\text{eff}}^{(H)} = \left(\sum_i \tilde{w}_i\right)^2 / \sum_i \tilde{w}_i^2$, $N_{\text{eff}}^{(A)} = \left(\sum_i \tilde{v}_i\right)^2 / \sum_i \tilde{v}_i^2$ quantify variance inflation. A generalized-Bayes weighted pseudo-posterior retains Beta-form updates with fractional sufficient statistics:

$$\theta_A \mid \mathcal{D} \sim \text{Beta}\left(\alpha_A + \tilde{n}_A,\ \beta_A + \tilde{N}_A - \tilde{n}_A\right),$$
$$\theta_{H|1} \mid \mathcal{D} \sim \text{Beta}\left(\alpha_1 + \tilde{n}_{11},\ \beta_1 + \tilde{n}_{10}\right),$$
$$\theta_{H|0} \mid \mathcal{D} \sim \text{Beta}\left(\alpha_0 + \tilde{n}_{01},\ \beta_0 + \tilde{n}_{00}\right).$$

Map draws to $g$ as usual. Because this is a weighted pseudo-posterior rather than the original Bernoulli likelihood, its calibration should be assessed separately. In practice, clip extreme weights and report $n_{\text{eff}}$.

## M   Reproducibility checklist

- Random seeds are fixed for the OOF, repeated-labeling, bootstrap, permutation, K-bin, and SBC analyses.

- The authoritative prediction table contains subject identifiers, OOF fold assignments, scan dates, scan-level ages, probabilities, and fixed-threshold decisions.

- The base and independent-bin K-bin analyses use direct conjugate sampling; no NUTS result is reported.

- SBC uses $M_{\text{SBC}} = 500$ datasets and $S = 1000$ posterior draws per dataset.

- Per-replication summaries and publication figures are generated from versioned CSV outputs.

## N   Code and data availability

An anonymized supplementary archive contains the code, scripts, configuration files, synthetic fixtures, automated tests, and summary-level reference results associated with the analyses in this paper. The synthetic experiments can be reproduced directly from the supplementary archive. Reproducing the ADNI case study additionally requires authorized access to the corresponding MRI data and metadata.

Participant-level ADNI data, subject identifiers, MRI files, metadata exports, and row-level prediction tables are not redistributed because they are governed by the ADNI Data Use Agreement. Qualified researchers may request access through `https://adni.loni.usc.edu/`. Accordingly, the supplementary archive contains the analysis code, data schemas, reproducibility instructions, and non-identifying summary-level results, but not participant-level ADNI data.

