# OpenReview forum: "Conjugate Bayesian Chain-Rule Prediction-Powered Inference for Binary Prevalence Estimation"
_TMLR — Under review for TMLR_

### Review · Reviewer_s2jB · 2026-06-20

**Summary Of Contributions:**

## Contributions

The authors study a specific case of Bayesian prediction-powered inference (PPI): estimating a binary prevalence functional via the chain rule. In this setting, when priors are independent Betas, the posterior factorizes into three independent Betas with closed-form updates, so uncertainty can be obtained by direct sampling without MCMC. The authors also characterize common use cases where conjugacy does not hold (e.g., K-bin scores, hierarchical pooling) and the No-U-Turn Sampler (NUTS) is truly needed. The authors also detail different metrics for auditing this chain-rule estimator (CRE) at deployment. Experiments compare CRE against 3 baselines - labeled-only Bayes, classical difference estimator, and prior-free analytic PPI estimator - and mainly on a single medical dataset, ADNI. CRE consistently shows better coverage and a narrower 95% interval width than the labeled-only Bayesian baseline at small label budgets, and neither CRE nor PPI dominates at the full cohort. The ADNI age-stratified case study further shows that CRE performs competitively with PPI even in a non-conjugate case.
***
## Strengths
1. The authors provide good practical motivation for studying this specific case of Bayesian PPI
2. The conjugacy result is clean, accurate, and applicable to many use cases.
3. The scoping of claims are detailed and accurate to what is presented in the paper.
4. The authors pay a lot of attention to details on formalizing the most common non-conjugate use cases and the practical deployment audits that would be essential for practitioners to adopt this method.
5. The baselines used to compare with CRE cover all classes of alternative methods that practitioners often consider.
6. Experiments are thorough in validating the theoretical claims as well as comparing the most relevant metrics used in practice (coverage and 95% interval width) and common use case for medical dataset (age-stratified)
***
## Weaknesses
1. It is unclear what advantage CRE offers over its strongest baseline, prior-free PPI. Their empirical performance closely aligns, and prior-free PPI is even used as a deployment-time audit for CRE. A clearer discussion of CRE's benefits and tradeoffs relative to prior-free PPI, whether theoretical or empirical (e.g., computational cost), would help justify CRE as a default method for binary prevalence monitoring with scarce labels.

**Audience:**

Yes

**Audience Explanation:**

Readers working on prediction-powered inference (PPI) and deployment monitoring of machine learning systems in label-scarce settings (e.g., clinical ML) would find the conjugate result and the deployment-audit toolkit useful and directly applicable.

**Broader Impact Concerns:**

A broader impact statement and ADNI data statement are present. There are no further concerns.

**Claims And Evidence:**

Yes

**Claims Explanation:**

The paper’s key claims are appropriately supported and carefully scoped throughout. Proposition 3.1 is a straightforward and correct consequence of Beta-Bernoulli conjugacy, and the simulation-based calibration experiment provides empirical validation of its accuracy. The repeated-labeling resampling study supports the claimed improvements of CRE over both labeled-only Bayesian estimation and the standard difference estimator. The ADNI case study examines one non-conjugate setting on a real dataset, consistent with the paper’s stated scope, and serves as a useful proof of concept for how CRE can be deployed and audited in practice. Experimental results are presented clearly with tables and figures, accompanied by accessible discussion of their implications and the practical advantages offered by CRE.

**Requested Changes:**

1. (Critical) Add a direct theoretical or empirical comparison clarifying CRE's advantage over prior-free PPI. An experiment isolating a setting with a measurable CRE advantage (e.g., accuracy, cost, derived functional) or a theoretical argument would help justify why CRE should be the default method for binary prevalence estimation. It would also be helpful to include this comparison in the abstract and introduction alongside the existing comparisons to labeled-only Bayesian baseline and the difference estimator.

---

### Review · Reviewer_1cNA · 2026-06-22

**Summary Of Contributions:**

This paper studies Bayesian prediction-powered inference for binary prevalence estimation when abundant machine predictions are available but human labels are scarce. The target prevalence is expressed through a chain-rule decomposition using the machine autorater output and the human label. The main contribution is to show that, under a Beta-Bernoulli specification, the posterior for the base model factorizes into independent Beta distributions, so posterior uncertainty for the prevalence functional can be obtained by direct sampling without MCMC.

The paper further discusses non-conjugate extensions, including hierarchical pooling, logit-normal priors, multi-bin score models, and threshold uncertainty, where NUTS may be useful. Empirically, the proposed conjugate chain-rule estimator is evaluated against labeled-only Bayes, a classical difference estimator, and an analytic PPI baseline. The experiments include simulation-based calibration, repeated-labeling resampling on ADNI-derived prediction tables, and an Alzheimer’s disease MRI case study with threshold and subgroup analyses. Overall, the paper presents a simple and practical Bayesian estimator for a focused deployment setting.

**Audience:**

Yes

**Audience Explanation:**

The paper should be of interest to at least some members of the TMLR audience, especially researchers working on prediction-powered inference, uncertainty quantification, semi-supervised inference, Bayesian methods, and reliable deployment of machine learning systems with limited labels.

The problem setting is practically relevant: many applications have abundant model predictions but limited human annotations, and the paper studies how to obtain calibrated prevalence estimates in such settings. The proposed conjugate chain-rule estimator is simple, computationally lightweight, and easy to audit, which may be useful for practitioners and researchers interested in deployment-time monitoring. The ADNI case study also provides a concrete example in medical imaging, where label scarcity and reliable uncertainty quantification are important concerns.

Overall, while the paper focuses on a specific binary prevalence estimation setting, its findings are likely to be relevant to a subset of the machine learning audience interested in statistically valid inference with machine-generated predictions.

**Broader Impact Concerns:**

I do not have major broader impact concerns. The paper studies uncertainty-aware prevalence estimation in label-scarce settings, with a medical imaging case study, and the authors already acknowledge the need for careful threshold selection, calibration, equity considerations, and domain governance. I think the existing broader impact discussion is sufficient for the scope of the paper. The authors may optionally add a brief note that the method should support monitoring and uncertainty quantification rather than replace clinical judgment or prospective validation.

**Claims And Evidence:**

Yes

**Claims Explanation:**

The main claims of the paper are supported by the presented derivations and experiments. The conjugacy result for the base binary chain-rule model follows clearly from the Beta-Bernoulli formulation, and the paper explains how posterior uncertainty for the prevalence functional can be propagated through direct sampling rather than MCMC. The simulation-based calibration experiment provides additional evidence that the conjugate posterior computation is behaving as expected under the assumed model.

The empirical results are also consistent with the paper’s stated claims. In the repeated-labeling resampling study on the ADNI-derived cohort, the proposed estimator achieves near-nominal coverage and narrower intervals than the labeled-only Bayesian baseline in the full cohort, especially at small label budgets. The comparison with the difference estimator and the analytic PPI baseline helps contextualize the benefits of the proposed approach. The ADNI case study further illustrates how the estimator can be combined with threshold selection, calibration checks, and subgroup analysis in a practical deployment setting.

Overall, the paper’s claims are clearly stated and generally supported by the evidence provided, particularly within the focused binary prevalence estimation setting considered by the authors.

**Requested Changes:**

I do not see major changes that are critical to the validity of the paper’s main claims. The following revisions would strengthen the submission and improve clarity.

1. Clarify the relationship to prior Bayesian PPI work. The paper already cites recent Bayesian formulations of PPI, but it would be helpful to make the distinction more explicit: what is new in this binary chain-rule specialization, what is inherited from existing Bayesian PPI formulations, and what practical advantage is gained by emphasizing the conjugate base case.

2. Better highlight the estimand distinction between the thresholded chain-rule estimator and the continuous-probability analytic PPI baseline. Since the baseline using continuous probabilities is competitive in several results, the paper should more clearly explain when the proposed thresholded CRE is preferable, for example in deployment settings where binary autorater decisions are the operational object of interest.

3. Provide more implementation detail for the out-of-fold prediction and threshold-selection pipeline. The ADNI case study is an important part of the paper, so the authors should make the data splitting, OOF score construction, threshold selection, and any calibration steps as transparent as possible to rule out ambiguity about possible leakage.

4. Expand the discussion of when the conjugate base model may fail. The paper already mentions covariate shift, label missingness, threshold uncertainty, and non-conjugate extensions. It would be useful to consolidate these into a clearer practical guide: when is the base CRE sufficient, and when should users move to hierarchical, multi-bin, importance-weighted, or threshold-uncertainty extensions?

5. Improve the presentation of the empirical comparison. In addition to reporting coverage and interval width separately, the paper could more explicitly summarize the coverage-width tradeoff across CRE, labeled-only Bayes, difference estimation, and analytic PPI. This would make the practical contribution easier to interpret.

6. Minor presentation improvements. Some parts of the paper repeat the same high-level motivation and limitation statements. The paper could be tightened by reducing repetition and moving some deployment details or extended discussions to the appendix.

---

### Review · Reviewer_KV4J · 2026-06-29

**Summary Of Contributions:**

The paper treats one specific instance of Bayesian prediction-powered inference: estimating binary prevalence g = θ_A·θ_{H|1} + (1−θ_A)·θ_{H|0}, where A is an abundant binary autorater and H a scarce human label. The central observation is that with independent Beta priors and Bernoulli likelihoods the base model's posterior factorizes into three independent Betas (Prop. 3.1), so uncertainty in g can be propagated by direct sampling instead of MCMC. The authors position this conjugate "CRE" as a lightweight default for deployment-time monitoring and reserve NUTS for the non-conjugate extensions (hierarchical pooling, logit-normal priors, K-bin models, joint threshold uncertainty). Evidence comes from simulation-based calibration, a repeated-labeling resampling study on ADNI-derived out-of-fold tables, and an ADNI MRI case study with thresholding and exchangeability diagnostics; the baselines are labeled-only Bayes, the difference estimator, and a prior-free continuous-score PPI estimator.

The conjugacy result is clean and practitioner-friendly, the conjugate-vs-MCMC split is a sensible way to organize the method, and the deployment audits (OOF-vs-leaky threshold selection, bootstrap cut-point dispersion, labeled/unlabeled propensity overlap) are a real and somewhat unusual strength.

**Audience:**

Yes

**Audience Explanation:**

The setting — many model predictions, few human labels, and a deployment estimand tied to a thresholded decision — is common, and the estimator is simple and cheap enough to actually use. Beyond the estimator, the audit steps (labeled/unlabeled overlap, leaky-threshold avoidance, threshold-variability quantification) are practically useful and transferable. Researchers in PPI, semi-supervised evaluation, and ML-assisted prevalence monitoring would find something to take away.

**Broader Impact Concerns:**

None.

**Claims And Evidence:**

No

**Claims Explanation:**

Citations. The three PPI-specific references all need attention, and one I could not verify at all.

- Guo & Lei (2021), "Confidence sets from prediction-powered inference," JASA, doi:10.1080/01621459.2021.1996374: I searched the title, the authors, and the bare DOI and found no matching record; the DOI does not resolve, and a 2021 date predates PPI (introduced in 2023). It is cited for PPI's finite-sample guarantees, so please supply a verifiable record or remove it; that claim is in any case covered by Angelopoulos et al. (2023).
- Angelopoulos et al. (2023): the author list, issue number, and DOI are wrong. The published article is Angelopoulos, Bates, Fannjiang, Jordan, and Zrnic, Science 382(6671):669–674, doi:10.1126/science.adi6000.
- Hofer et al. (2024): arXiv:2405.06034 is "Bayesian Prediction-Powered Inference" by Hofer, Maynez, Dhingra, Fisch, Globerson, and Cohen — not the title and authors listed. Since the paper positions itself against this work, please state precisely what is new relative to it; Hofer et al. explicitly treat discrete autoraters, so the conjugate binary case may overlap and the delta should be made explicit.

All three have an incorrect author list, so I'd recommend checking the full bibliography against primary sources. None of this affects the conjugacy derivation, but it affects positioning and scholarly accuracy.

"PPI does not dominate CRE." I had trouble squaring this (§4.1) with Table 1. The prior-free PPI intervals are narrower than CRE at every budget while holding near-nominal coverage — e.g. at n=20, PPI is 0.966/0.246 versus CRE 0.980/0.280 (Uniform) and 0.972/0.264 (Jeffreys), with the same pattern at n=10, 40, 80. CRE clearly beats labeled-only Bayes and is far more stable than the difference estimator at small n, but on this evidence continuous-score PPI is at least competitive and usually narrower. I'd reframe the claim: CRE gives a coherent Bayesian posterior for the thresholded estimand, improves on labeled-only Bayes and the unstable difference estimator, and is empirically compatible with PPI — rather than stating that PPI does not dominate it.

Coverage wording. "Near-nominal" fits most rows, but the n=10 Uniform-prior CRE result (0.990) is better described as conservative. Worth noting that the Uniform prior can overcover at the smallest budget.

Figure 7. The vertical line at g_true = 0.60 doesn't match either reported target (0.308129 for the full cohort, 0.225410 for the 65–70 subset). If this is a separate synthetic diagnostic, please say so and explain the 0.60; otherwise relabel or remove it.

Proposition A.1. The O_P(N_A^{-1}) equivalence to the difference estimator needs justification. The gap between the chain-rule plug-in and ĝ_diff appears to involve the full-sample versus labeled-sample autorater means, and since the labeled set is the smaller component I'd expect a discrepancy governed by N_H rather than N_A^{-1} without extra conditioning. Please tighten the derivation or state the assumptions, or revise to a more conservative statement.

Case study. Please define H explicitly (AD vs CN? impaired vs CN?) — with prevalence ≈ 0.308 this should be one sentence. And please confirm the OOF folds are grouped at the subject level rather than the scan/visit level: an AUC ≈ 0.97 from a two-block CNN trained five epochs is high enough that ruling out within-subject leakage matters for the case study's credibility. This does not bear on the CRE method itself.

Extensions. The abstract and contributions list hierarchical pooling, logit-normal priors, K-bin models, and joint threshold uncertainty, but only the conjugate base case is really exercised (K-bin gets a small sensitivity table). The Limitations section is honest about this. Either align the framing with what is shown, or include one end-to-end non-conjugate run with the usual NUTS diagnostics.

What holds up. Prop. 3.1 is correct; the SBC study is the right check on the inference engine and looks fine; the resampling tables are internally consistent. The well-supported claims are that CRE is narrower than labeled-only Bayes in the full cohort, much more stable than the difference estimator at small n, and competitive but with attenuated gains in the 65–70 subset.

**Requested Changes:**

1. [Critical] Verify and correct the PPI references: replace or remove Guo & Lei (2021), which I could not verify exists; fix the author list, issue, and DOI for Angelopoulos et al. (2023); fix the title and authors for Hofer et al. (2024). Given all three are wrong, audit the full bibliography.
2. [Critical] Clarify the relationship to Hofer et al. (2024), particularly given their discrete-autorater treatment. A clean specialization plus operational evaluation is a fine contribution, but the delta should be stated.
3. [Critical] Revise or justify the claim that PPI "does not dominate CRE" (§4.1); Table 1 shows PPI narrower at every budget at near-nominal coverage.
4. [Critical] Resolve Figure 7 — g_true = 0.60 does not match the reported targets.
5. [Critical] Clearly define H in the case study and confirm subject-level OOF grouping.
6. [Critical] Provide a fuller derivation of the O_P(N_A^{-1}) claim in Prop. A.1, or weaken the statement.
7. [Strengthen] Consider adding PPI++ (Angelopoulos, Duchi, Zrnic, arXiv:2311.01453) as a comparison, or justify the vanilla baseline. The analytic PPI used here is the λ=1 case of PPI++; since PPI is already narrower than CRE, the variance-tuned version would only widen that gap, so this bears directly on item 3. Optionally, also plot the model-only imputed estimate as a reference line to make the bias–variance picture explicit.
8. [Strengthen] Align the abstract/contributions with the demonstrated extensions, or add one end-to-end non-conjugate example with NUTS diagnostics (R̂, bulk/tail ESS, divergences, E-BFMI).
9. [Strengthen] Use one name for the labeled-only Bayesian baseline ("NB"/"labeled-only Bayes" in the text versus "Naïve" in the tables).
10. [Strengthen] Consider replacing "near-nominal" with "conservative-to-near-nominal" where the Uniform prior overcovers (n=10).

---

### Author Response · Authors · 2026-07-12
**Substantial revision uploaded**

We thank the reviewers and the Action Editor for their detailed and constructive feedback.

We have now uploaded a substantially revised manuscript together with an updated anonymized supplementary archive. The revision addresses the main points raised in all three reviews, including the correction and audit of the PPI bibliography, clarification of the relationship to prior Bayesian PPI work, a revised discussion of CRE relative to continuous-score PPI and the power-tuned baseline, subject-grouped out-of-fold prediction, the updated age 65--70 analysis, revised coverage and interval-width results, additional threshold and calibration diagnostics, and the corrected appendix derivation relating the chain-rule plug-in estimator to the binary difference estimator.

A detailed summary of the changes is provided in the “Changes Since Last Submission” field.

We have a final discussion with the co-author scheduled for July 16. The current upload contains the substantive revision and can be reviewed now; following that discussion, we may upload a further revision changes if needed. We will post an additional comment if a second revision is uploaded.

We would be grateful if the reviewers could consider the revised manuscript and supplementary material when preparing their recommendations.

---

### Author Response · Authors · 2026-07-16
**Final co-author review completed**

We thank the reviewers and the Action Editor again for their careful and constructive feedback.

We have now completed the final co-author review of the revised manuscript and supplementary material. The current OpenReview revision contains the substantive changes addressing all three reviews, and we confirm that it represents our final revised version at this stage.

In particular, the revision includes the corrected and audited PPI bibliography, clarified positioning relative to prior Bayesian PPI work, the revised CRE-versus-continuous-score PPI comparison, the power-tuned PPI++-style baseline, subject-grouped out-of-fold prediction, scan-level age matching and the updated age 65--70 analysis, revised coverage and interval-width interpretations, additional deployment diagnostics, and the corrected appendix derivation.

We do not currently plan any further revisions unless the reviewers or the Action Editor request additional clarification. We would be grateful if the reviewers could consider the revised manuscript and anonymized supplementary material when preparing their recommendations.